


# The future of the El Niño-Southern Oscillation: Using large ensembles to illuminate time-varying responses and inter-model differences

Nicola Maher[1,2], Robert C. Jnglin Wills[3], Pedro DiNezio[2], Jeremy Klavans[2], Sebastian Milinski[4,5], Sara C. Sanchez[2], Samantha Stevenson[6], Malte F. Stuecker[7], and Xian Wu[8]

[1]Cooperative Institute for Research in Environmental Sciences (CIRES), University of Colorado at Boulder, Boulder, CO 80309, USA
[2]Department of Atmospheric and Oceanic Sciences (ATOC), University of Colorado at Boulder, Boulder, CO 80309, USA
[3]Department of Atmospheric Sciences, University of Washington, Seattle, WA 98195, USA
[4] Climate and Global Dynamics Division, National Center for Atmospheric Research, Boulder, CO 80307, USA
[5]Cooperative Programs for the Advancement of Earth System Science, University Corporation for Atmospheric Research, Boulder, CO 80307, USA
[6]Bren School of Environmental Science and Management, University of California, Santa Barbara, Santa Barbara, CA 93106, USA
[7]Department of Oceanography  International Pacific Research Center (IPRC), School of Ocean and Earth Science and Technology (SOEST), University of Hawai'i at Mānoa, Honolulu, HI, USA
[8]Climate and Global Dynamics Division, National Center for Atmospheric Research, Boulder, CO 80305, USA

**Correspondence:** Nicola Maher (nicola.maher@colorado.edu)

**Abstract.** Future changes in the El Niño–Southern Oscillation (ENSO) are uncertain, both because future projections differ between climate models and because the large internal variability of ENSO clouds the diagnosis of forced changes in observations and individual climate model simulations. By leveraging 14 single model initial-condition large ensembles (SMILEs), we robustly isolate the time evolving response of ENSO sea surface temperature (SST) variability to anthropogenic forcing from internal variability in each SMILE. We find non-linear changes in time in many models and considerable inter-model differences in projected changes in ENSO and the mean-state tropical Pacific zonal SST gradient. We demonstrate a linear relationship between the change in ENSO SST variability and the tropical Pacific zonal SST gradient although forced changes in the tropical Pacific SST gradient often occur later in the 21st century than changes in ENSO SST variability, which can lead to departures from the linear relationship. Single forcing SMILEs show a potential contribution of anthropogenic forcing (aerosols and greenhouse gases) to historical changes in ENSO SST variability, while the observed historical strengthening of the tropical Pacific SST gradient sits on the edge of the model spread for those models for which single forcing SMILEs are available. Our results highlight the value of SMILEs for investigating time-dependent forced responses and inter-model differences in ENSO projections. The non-linear changes in ENSO SST variability found in many models demonstrate the importance of characterising this time-dependent behaviour, as it implies that ENSO impacts may vary dramatically throughout the 21st century.



# 1   Introduction

Understanding how the El Niño–Southern Oscillation (ENSO) will change under increasing greenhouse gas emissions is critical due to ENSO's widespread impacts, which include changes to floods, droughts, and fisheries production (e.g. McPhaden et al., 2006; Taschetto et al., 2020; Cai et al., 2021). While previous work demonstrates model agreement on future intensification of ENSO's atmospheric impacts (e.g. precipitation; Yun et al., 2021; Power et al., 2013; Fasullo et al., 2018), there remains debate as to whether ENSO-driven sea surface temperature (SST) variability will increase or decrease in the future as disparities between projections from different climate models persist (e.g. Cai et al., 2022; Wengel et al., 2021; Beobide-Arsuaga et al., 2021; Stevenson et al., 2021). In this study we present ENSO SST projections for the first time from 14 individual single model initial-condition large ensembles (SMILEs) where the time-evolving forced response is cleanly separated from internal variability.

Previous work finds a diverse range of projections of ENSO variability in multi-model ensembles (CMIP; coupled model intercomparison projects). In CMIP3 a range of ENSO variability projections were found leading to a review stating that it was not yet possible to determine whether ENSO would change in the future (Collins et al., 2010). In CMIP5 the models again demonstrate a variety of responses with the multi-model mean change indistinguishable from zero (Stevenson, 2012; Beobide-Arsuaga et al., 2021). Other studies have further investigated ENSO projections by excluding models that have strong ENSO biases in the historical period. When selecting for models that correctly represent ENSO skewness, Cai et al. (2018) find an increase in Eastern Pacific ENSO SST variability. This result is supported by the recent release of CMIP6 wherein Cai et al. (2022) demonstrate that 34 of 43 models show an increase in ENSO SST variability in SSP585 (27 of 39 in SSP126) in agreement with another early CMIP6 assessment (Fredriksen et al., 2020).

There is also some evidence for an increase in ENSO variability in the observational record, with paleoclimate data suggesting that ENSO variability strengthened post-1950 (Cai et al., 2021). Coral records indicate that ENSO variability has been 25% larger over the last 5 decades compared to the pre-industrial (Grothe et al., 2020; Cobb et al., 2013). This agrees with tree ring records, which show a strengthening of ENSO variance in the late 20th Century (Li et al., 2013). A study using linear inverse modelling based on instrumental data agrees that ENSO SST variance has increased since 1976 (Capotondi and Sardeshmukh, 2017). The authors, however, note that such differences could be due to multi-decadal variability or to the impact of climate change, and the cause of this observed change remains unclear.

While consensus between multi-model ensembles and paleoclimate records suggests that ENSO variability may increase with increasing greenhouse gas emissions, other studies contest this result. A recent study using a very high-resolution climate model (0.1 degree ocean; 0.25 degree atmosphere) finds that ENSO SST variability decreases substantially under strong ($4xCO_2$) forcing (Wengel et al., 2021). This study further argues that CMIP class models are too low in horizontal resolution to correctly capture some important aspects of ENSO ocean dynamics, which could lead to an artificial increase in SST variance in these models. On longer time-scales, ENSO amplitude is also found to decrease in CMIP class standard (1 to 2 degree) resolution models (Callahan et al., 2021). Using output from the Long Run Model Intercomparison Project, Callahan et al. (2021) find diverging transient ENSO variability projections, but a consistent decrease in SST variability across models





when the response has stabilized. Indeed Kim et al. (2014) suggest that ENSO variability in CMIP5 models is not linear in time, implying an important role for analysing multiple time periods. Collectively, these studies demonstrate the importance of further research to reconcile these discrepancies.

Differences between the projections of ENSO variability in individual models have been linked to the pattern of mean-state warming across the tropical Pacific (e.g. Jin and Neelin, 1993; Fedorov and Philander, 2000; Jin et al., 2006; DiNezio et al.,
2012). For instance, in CMIP5/6 models there is a weak negative correlation between the magnitude of the tropical Pacific SST gradient and ENSO SST variability (Beobide-Arsuaga et al., 2021). In this case, an increase in ENSO SST variability is linked to a weakening of the tropical Pacific SST gradient. This relationship varies between models (Fredriksen et al., 2020) and is found to be more robust in models that have more realistic subsurface nonlinear dynamical heating and ENSO asymmetry when compared to observations (Hayashi et al., 2020). Periods of weaker tropical Pacific SST gradient are also linked to higher
ENSO variability, with periods with a stronger gradient linked to periods of smaller ENSO SST variability (Capotondi and Sardeshmukh, 2017; Rodgers et al., 2004; Ogata et al., 2013; Choi et al., 2013; McPhaden et al., 2011). Models that show reduced ENSO variability in response to anthropogenic forcing also show a strengthening of the tropical Pacific SST gradient (Kohyama and Hartmann, 2017; Kohyama et al., 2017). Additionally, the relationship between ENSO amplitude and the mean-state tropical Pacific SST gradient is identified in PMIP models, although it is not constant through time (Wyman et al., 2020).
We emphasize that east-west Pacific SST gradient is only one of many mean state metrics that are potentially important to influence (and vice versa be influenced by) ENSO variability. In fact, different aspects of the climate mean state affect ENSO feedbacks in multiple ways, leading to a potentially high sensitivity of the linear ENSO growth rate (and hence SST variability) to these factors (Jin et al., 2006).

How the tropical Pacific SST gradient itself will change in the future is also a point of contention. Most CMIP class models
agree on a projected weakening of the SST gradient (El Niño-like warming; Kociuba and Power, 2015; Meehl and Washington, 1996; Fredriksen et al., 2020; Cai et al., 2021). This, however, must be reconciled with the recent observed increase in the SST gradient (La Niña-like warming; Kosaka and Xie, 2013; England et al., 2014; McGregor et al., 2014). Some studies suggest that the inconsistencies between models and observations can be explained by internal variability and observational uncertainty (Watanabe et al., 2021; Chung et al., 2019; Coats and Karnauskas, 2017), while others argue that the observations are truly
outside the model range (Kociuba and Power, 2015; Seager et al., 2022). One hypothesis is that under transient forcing the ocean thermostat mechanism, where upwelling waters delay warming in the eastern equatorial Pacific, overwhelms the long-term weakening of the Walker circulation and SST gradient (Clement et al., 1996; Heede and Fedorov, 2021), leading to temporary La-Niña-like warming. Another study concludes that a more accurate representation of ENSO nonlinearity leads to La Niña-like warming in at least one realistic climate model (Kohyama and Hartmann, 2017; Kohyama et al., 2017). As
such, while most models project an El Niño-like warming, this is not the only possible future response. Alternatively, realistic ENSO nonlinearity does not necessarily lead to a La Niña-like warming pattern. While realistic representations of ENSO nonlinearities are a necessary condition for the simulation of rectified mean state changes via nonlinear dynamical heating, diverging projected changes in future ENSO SST variability among models with largely realistic ENSO nonlinear dynamic heating can lead to either ENSO-induced damped or amplified eastern tropical Pacific mean-state warming (Hayashi et al.,





2020). Additionally, it has been hypothesized that SST mean state biases in the Atlantic Ocean are responsible for low biased interbasin connectivity (and hence also low-biased SST variability) between the Atlantic and Pacific (McGregor et al., 2018). This suite of studies highlights the need for further research and continued observations to reduce uncertainties in how the tropical Pacific SST gradient will change in the future.

A primary reason for uncertainty in future changes in ENSO variability and the mean-state tropical Pacific SST gradient is
that ENSO itself experiences large multi-decadal variability (Wittenberg, 2009), meaning that long averaging periods or large ensembles are needed to identify robust forced changes (e.g. Stevenson et al., 2010; Milinski et al., 2020; Maher et al., 2018; Deser et al., 2020; Ng et al., 2021). Projections using multi-model ensembles can be difficult to interpret because the spread due to internal variability is comparable to the spread due to inter-model differences (Maher et al., 2018; Ng et al., 2021). With this in mind, large ensembles are needed to quantify the time-varying ENSO response (Stevenson, 2012; Milinski et al., 2020).
Without large ensembles, previous studies were forced to rely on long time averages to robustly evaluate changes in ENSO variability (e.g., 2000-2099 compared to 1900-1999).

In this study, we present ENSO projections from 14 SMILEs, where the time-dependent response of ENSO to external forcing can be isolated from internal variability through ensemble averaging. The unprecedented number of SMILEs used allows a detailed examination of the model-dependence of future ENSO projections. In particular, this paper aims to:

– Isolate the forced response of ENSO in individual models

– Evaluate the time evolution of projected changes in ENSO statistics

– Investigate the time-dependent mean-state response of the tropical Pacific (tropical Pacific SST gradient) to greenhouse gas forcing

– Determine whether there is a relationship between projected changes in ENSO variability and the tropical Pacific SST
gradient in climate models

– Investigate the role of greenhouse gases, aerosol, and natural forcing in historical ENSO variability changes using single forcing SMILEs

## 2  Datasets

The 14 SMILEs used in this study include both CMIP5 and CMIP6 class models and use one of three external forcing scenarios
(Table 1). While the purpose of this study is to provide a detailed overview of ENSO projections in all available SMILEs, it is useful to provide some observational context. We note that a detailed comparison of ENSO characteristics in both CMIP models and SMILEs can be found in the following studies for a large range of ENSO metrics, and we do not repeat this here (Planton et al., 2021; Lee et al., 2021). We do, however, compare model-simulated and observed El Niño and La Niña composites (Figure S1 & S2). We find diverse errors in ENSO patterns, amplitude, and transitions. All models overestimate the



westward extension of ENSO SST anomalies and either overestimate/underestimate the ENSO amplitude, and some models
overestimate the duration of ENSO events (e.g., MIROC6  MIROC-ES2L).

When considering ENSO SST variability we find that only some models capture the observed variability within the ensemble
spread (Figure 1), but note that this is not a reliable predictor for future change. For example, both MPI-GE and EC-EARTH3
compare favourably with the observed magnitude of ENSO SST variability, but have different future responses (Figure 1).

MPI-GE shows no change in ENSO SST variability, while EC-EARTH3 has strongly increasing ENSO SST variability. We
additionally note that there is large multi-decadal variability in the amplitude of ENSO variability in individual ensemble
members. This indicates that we need a SMILE to robustly detect a forced change, but also that an actual forced change in the
real world might be masked by this large multi-decadal variability.

## 3  ENSO Projections

In this section we evaluate the response of ENSO variability in each individual SMILE in one of three future scenarios depen-
dent on availability (RCP8.5, SSP370, SSP585).

### 3.1  Amplitude

Projections of ENSO amplitude are non-linear in time and differ between models (Figure 1). A large proportion of the mod-
els demonstrate an increase in ENSO amplitude as characterized by December-January-February (DJF) SST variability of

three ENSO indices (Niño3, Niño3.4  Niño4; ACCESS-ESM1.5, CESM1-LE, CSIRO-Mk36, EC-EARTH, IPSL-CM6A-LR,
MIROC6, MIROC-ES2L). This increase is often non-linear in time, and there is a large spread across models in the magnitude
of the increase. Other models show a decrease (CanESM2) or limited change (MPI-GE) in ENSO amplitude. Finally, five
models display non-monotonic behaviour (CESM2-LE, GFDL-CM3, GFDL-ESM2M, CanESM5, GFDL-SPEAR-MED). In
these models, ENSO amplitude first increases, then plateaus, and lastly decreases. These results are consistent for all three

ENSO indices, with subtle differences for some models (e.g., CSIRO-Mk36  IPSL-CM6A-LR). Given the time-dependent re-
sponse of ENSO amplitude exhibited in these time-series plots, we consider projections for two future time periods (2021-2050
2070-2099) in the following sections.

### 3.2  Pattern

Forced changes in the spatial pattern of ENSO-related SST variability differ between models (Figures 2 & S3). While the multi-

ensemble mean (MEM) shows an increase in variability in the central Pacific, each model has its own unique pattern of change
(Figure 2). Some models demonstrate a general decrease in SST variability along the equator (CanESM2, CESM2, MPI-GE)
while others show a general increase in this region (CESM1, EC-EARTH, MIROC6, MIROC-ESL2L, GFDL-SPEAR-MED).
There are also differences in the zonal pattern of variability changes. A group of models has large increases in variability in the
central Pacific (ACCESS-ESM1.5, CanESM5, IPSL-CM6A-LR), while another group shows a clear increase across the Pacific

(CESM1-LE, EC-EARTH, GFDL-SPEAR-MED, MIROC-ES2L). These diverse patterns are generally similar between the two





time periods considered, but smaller in magnitude for the earlier period (Figure S1; 2021-2050) compared to the later period (Figure 2; 2071-2100), with the MEM pattern consistent between time periods. This result is, however, not true for all models. CESM2 has a central equatorial Pacific response of opposite sign between the two periods considered, while GFDL-CM3 and CSIRO-Mk36 have very different patterns of spatial change in the two periods considered. These figures give the pattern of
overall change in ENSO SST variability, however, El Niño and La Niña are not symmetric and may evolve in different ways. Due to this we next consider changes in El Niño and La Niña events individually.

### 3.3 ENSO Event Evolution

Composites of El Niño events in each period show increased El Niño SST amplitude relative to the baseline period 1951-1980 in most models and the MEM (Figure 3; 2070-2099, Figure S4; 2021-2050). In the baseline period, most models have
maximum SST anomalies in the eastern equatorial Pacific sometime between November and February followed by a westward propagation of warm anomalies and then a switch to cold anomalies in the following year. In the MEM, the longitudinal location of the maximum SST anomaly shifts westward by approximately $15^{o}$ in 2070-2099 compared to that in 1951-1980, indicating that El Niño SST variability gets stronger over the central Pacific in the future period, consistent with results shown in Figure 2. In models where the amplitude of El Niño increases, the amplitude of the subsequent La Niña also tends to increase
(Figures S5 & S6), presumably due to a stronger heat content discharge, and vica versa for models where the amplitude of El Niño decreases. When considering the asymmetry between El Niño and La Niña event composites, the MEM shows that the amplitude of La Niña intensifies more than El Niño in the western-central equatorial Pacific, and the duration of La Niña becomes longer following the amplitude intensification, highlighting the non-linearity of their responses (Figure S5). The individual model responses, however, demonstrate strong inter-model differences, which might be related to the diverse model biases in simulating ENSO (Figures S1 & S2). Overall, ENSO SST changes in the period 2021-2050 show similar patterns,
albeit weaker amplitude, to those in the periods in 2070-2099 (Figures S4 & S6). CESM2 is an exception that has opposite changes in El Niño SST amplitude and La Niña duration between the two periods.

### 3.4 Seasonal Synchronisation

We next investigate the seasonal synchronization of projected changes in ENSO SST variability (Figures 4 & 5). We aim to
determine during which seasons ENSO SST variability changes in each SMILE and determine whether the ENSO seasonality is likely to increase or decrease. We find that models demonstrate different magnitudes of change for different months (Figure 4). Most models show the largest change (generally an increase) in boreal winter with limited changes in boreal summer. Individual models have opposite signed changes between the two seasons. Overall, there is an increase in ENSO seasonality (Figure 5), i.e., increases in ENSO variability are concentrated in the season where ENSO SST variability is climatologically
largest. These changes are largely consistent across models, with an overall increase in ENSO SST variability found in boreal fall and winter and a decrease in the variability in spring and summer.





## 4 Mean-state projections

The warming rate over the equatorial Pacific varies seasonally and as a function of longitude (Figure 6). While some models have larger warming trends in the east Pacific (El-Niño-like warming) for all months (MIROC6, CESM1, EC-EARTH, CESM2), others demonstrate a more complicated seasonal cycle of warming rate, with the location of largest warming dependent on which month is considered. In the MEM, the eastern Pacific warms more in all seasons, however, from December to June the strong warming trend extends into the central Pacific, which warms with almost equal magnitude to the east. The MEM warming rate is larger over the eastern equatorial Pacific during late boreal spring (the climatological warm season of the equatorial Pacific) compared to boreal autumn (the cold season), but the season of strongest warming is model-dependent. For example, GFDL-CM3 has the largest eastern equatorial Pacific warming in boreal summer, while CESM1 has the largest warming in boreal fall and winter.

The time evolution of tropical Pacific SST gradient changes is investigated by considering the SST difference between the eastern and western equatorial Pacific (Figure 7). A decrease in this gradient indicates stronger warming in the east Pacific, i.e., El Niño-like warming (red colours in Figure 7), while a strengthening of the gradient corresponds to La Niña-like warming (blue colours in Figure 7). We find El Niño-like warming in most models, which is evident in the MEM. This is, however, not the case for CSIRO-Mk36 and is seasonally dependent for GFDL-CM3, GFDL-ESM2M, IPSL-CM5A-LR, and MPI-GE. The season during which projected changes are strongest is again model dependent, but SST gradient changes are often concentrated in the season during which the climatological SST gradient is strongest in the specific model considered. The following section relates the projected changes in ENSO variability to the changes in the tropical Pacific SST gradient to investigate whether these changes are linked.

## 5 Relationship between ENSO variability and mean-state changes

The projected change in ENSO SST variability is linearly related to the projected change in the tropical Pacific SST gradient across ensemble members from all models in the period 2021-2050 (Figure 8). This relationship breaks down in the later period (2070-2099) for models that have large SST gradient changes (e.g. CESM2; CSIRO-Mk3.6; CanESM2). In general, an ensemble member that experiences El Niño-like warming is most likely to see an increase in ENSO variability, while a La Niña-like warming is associated with the opposite response. While this does not occur for all ensemble members, the larger the response in either the tropical Pacific SST gradient or ENSO SST variability the more likely that a concurrent change in the other variable will be seen.

We further investigate the relationship between changes in ENSO SST variability and the change in the tropical Pacific SST gradient by plotting the evolving relationship between the two variables over time (Figure 9). Many models have a hook-shaped ensemble-mean (forced) response that corresponds to the following time-dependent response. First, there is an increase in ENSO SST variability concurrent with uniform warming across the Pacific (i.e., no change in tropical Pacific SST gradient) or weak El Niño-like warming. Then there is El Niño-like warming concurrent with a plateau in ENSO SST variability. In some models, this is followed by a decrease in ENSO SST variability as the mean state continues to warm in an El Niño-like fashion.





This behavior is seen in seven models (CESM1-LE, GFDL-SPEAR, MIROC6, MIROC-ES2L, CanESM5, CESM2-LE, IPSL-CM6A-LR), with an additional four that demonstrate a portion of the hook-shaped response (EC-EARTH3, ACCESS-ESM1-6, CanESM2, MPI-GE). The three outliers are GFDL-ESM2M, which exhibits an increase then a decrease in both ENSO SST variability and the mean-state tropical Pacific SST gradient, GFDL-CM3, which shows an increase then a decrease in ENSO SST variability without much change in the mean-state tropical Pacific SST gradient, and CSIRO-Mk36, which has strong La

Niña-like warming and an increase in ENSO SST variability.

The time evolution of ENSO SST variability and the tropical Pacific SST gradient (Figure 9) provide context for the results in Figure 8. While projections of ENSO variability and the tropical Pacific SST gradient are linearly related as shown in Figure 8, some models show a forced response with the opposite relationship, and the time evolution reveals that ENSO variability changes precede the mean-state SST gradient changes in most models, supportive of the hypothesis that ENSO nonlinear

dynamical heating can rectify into the mean state (e.g. Hayashi et al., 2020). This highlights the value of using SMILEs to analyse the time-dependent response of the tropical Pacific in climate models. We additionally note that the individual ensemble member spread is much larger for ENSO SST variability projections than the tropical Pacific SST gradient (Figure 9). This means that more ensemble members are needed to retrieve the forced change in ENSO SST variability under warming (as well as higher order moments of other Earth system variables; Rodgers et al., 2021), but somewhat fewer are needed to

isolate the forced change in the tropical Pacific SST gradient (e.g. Wills et al., 2020; Rodgers et al., 2021).

## 6   A single forcing perspective

Using two SMILEs (CanESM5 and MIROC6) that have single forcing simulations available, we separate the contribution of aerosols, greenhouse gases, and natural forcing to historical changes in both ENSO SST variability and the tropical Pacific SST gradient (Figure 10). Both models have forced increases in ENSO variability over this time period (1985-2014 compared

to 1951-1980) although this is not evident in all ensemble members. This increase is consistent with observations and is driven by anthropogenic forcing (greenhouse gases and aerosols), not natural forcing. The larger ensemble (MIROC6) has strong variability across ensemble members in the natural forcing ensemble, where large increases or decreases could be observed in a single realization (or observations) due to this variability alone. This indicates that ENSO SST variability changes have not yet emerged from internal variability, consistent with previous work (e.g. Ying et al., 2022).

Over the historical period, the tropical Pacific SST gradient weakens in MIROC6 and does not change in CanESM5 in response to all forcings. This is a result of the combination of greenhouse gas and aerosol forcing, which generally oppose each other (Figure 10). Specifically, greenhouse gases act to weaken the gradient in both models, while aerosols strengthen the gradient in CanESM2 and have minimal effect in MIROC6. The observed change in the tropical Pacific SST gradient sits right at the edge of the ensemble spread in the all-forcing historical scenario. This suggests that the observed gradient could be due

to internal variability, however, it would be an extreme event. Differences between these models and observations could also be due to an incorrect model response to external forcing.





## 7 Discussion

While the MEM change in ENSO SST variability in this study compares well with previous literature, the SMILEs add information by allowing for the isolation of the time-dependent ENSO response in individual models. In most models, although not all, ENSO amplitude is projected to increase, in agreement with Cai et al. (2018, 2022); Fredriksen et al. (2020). However, while previous work used the differences between long time periods (i.e., 2000-2099 compared to 1900-1999) to isolate the projected change, we leveraged SMILEs to separate the time-dependent response of ENSO amplitude to external forcing from internal variability. Individual SMILEs have a range of projected evolutions of ENSO amplitude that are not linear in time. Additionally, in CESM2-LE, GFDL-CM3, and GFDL-ESM2M ENSO amplitude first increases, then decreases. In these models, the response captured using long time averaging is dependent on the specific period used. Here, SMILEs add value as an important new tool to fully capture the time-dependent ENSO variability projections.

The spatial pattern of projected changes in ENSO variability also differs between models. In the MEM, we find a general shift toward more variability in the central Pacific. However, the individual model patterns that make up the MEM are diverse, and more work is needed to understand the model differences that we find. The MEM increase in the central Pacific variability is also clear in the MEM composite of El Niño event evolution. Additionally, La Niña events appear to strengthen in the MEM more than El Niño events, highlighting non-linear behaviour. In general, for both pattern and event evolution changes, the individual model responses are similar between the two time periods considered (2021-2050 2070-2099), albeit smaller in magnitude for the earlier period. However, this is not true for all models, with CESM2-LE a clear outlier showing opposite changes in the earlier and later periods.

We find that the seasonal synchronisation of ENSO SST variability increases in most models investigated in this study. This manifests in an increase in ENSO SST variability in boreal fall and winter and a decrease in spring and early summer. These changes have implications for understanding ENSO's intrinsic dynamics (e.g. Stuecker et al., 2013; Stein et al., 2014; Chen and Jin, 2022), changes to remote ENSO impacts (e.g., potentially amplifying the ENSO impact in one season for a given region and reducing it in a different season), and potential changes to ENSO predictability (e.g. Balmaseda et al., 1995; Timmermann et al., 2018). Thus, changes in variability should be assessed seasonally and not by using annual averages.

The tropical Pacific SST gradient response also agrees with previous work showing an El Niño-like warming in most models and the MEM (e.g Kociuba and Power, 2015; Fredriksen et al., 2020; Lian et al., 2018; Cai et al., 2021). However, again the individual model responses that go into the MEM are diverse, with the warming varying both seasonally and longitudinally. In general, the season of strongest SST gradient weakens the most, however, this is again not consistent across all models. In most models, weakening of the SST gradient intensifies in the second half of the twentieth century, consistent with a weakening of the Walker circulation gradually overwhelming the ocean thermostat mechanism (Heede et al., 2020; Heede and Fedorov, 2021).

Similar to the previous hypotheses, we find a link between the projected tropical Pacific SST gradient and the change in ENSO SST variability (DiNezio et al., 2012; Beobide-Arsuaga et al., 2021; Fredriksen et al., 2020; Hayashi et al., 2020; Wyman et al., 2020; Capotondi and Sardeshmukh, 2017; Rodgers et al., 2004; Ogata et al., 2013; Choi et al., 2013). However,

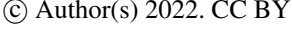



the SMILEs provide a much clearer picture of the time evolution of anomalies. We find a large subset of models that have an increase in ENSO SST variability that precedes the mean state change, with ENSO SST variability plateauing, and in some models decreasing as the mean state warms in an El Niño-like fashion. This result begins to conceptually link together previous work that found an increase in ENSO SST variability over the coming century (e.g. Cai et al., 2022) and a decrease in

the longer-term equilibrated state (e.g. Callahan et al., 2021). The individual model responses may also put into context a recent high-resolution study (Wengel et al., 2021) that finds a decrease in ENSO SST variability and argues that CMIP class models cannot capture all aspects of ENSO ocean dynamics correctly, leading to a potentially incorrect projected increase in ENSO SST variability. Based on our study, their model could be capturing a longer-term decrease in ENSO SST variability due to the strong forcing used ($4xCO_2$). Alternatively, we do find CMIP class SMILEs that show a decrease in ENSO SST variability,

highlighting the need to use multiple models for robustness. Based on these results, high-resolution modelling studies that use multiple models are needed to reconcile projections and determine the robustness of the higher-resolution result as compared to standard resolution CMIP class models.

In the two models that have single forcing SMILE experiments, greenhouse gas and aerosol forcing contribute to a historical increase in ENSO SST variability, however, greenhouse gas and aerosol forcing have competing influences on historical trends

in the tropical Pacific SST gradient. The observed change in the tropical Pacific SST gradient sits right at the edge of all ensemble members of the all-forcing historical scenario (consistent with McGregor et al., 2014; Seager et al., 2019, 2022; Wills et al., submitted 2022). This suggests that the modelled SST gradient response to greenhouse gas forcing could be too strong or incorrect, the modelled SST gradient response to aerosol forcing could be too weak, the observed change could be an extreme event caused by internal variability, or the modelled internal variability could be too small, or some combination of

the above. Further investigation is needed as more single forcing ensembles are released to determine the relative roles of these possibilities.

## 8   Summary and Conclusions

In this study we highlight the value of SMILEs for investigating ENSO projections. SMILEs allow the isolation of the time-dependent response of ENSO SST variability to increasing greenhouse gases. By using SMILEs we can better understand

projected changes in ENSO variability and the tropical Pacific SST gradient and isolate inter-model differences in their response to anthropogenic forcing.

Our results show the following:

1. ENSO SST Projections

    (a) Projections of ENSO amplitude are not linear in time.

(b) Models differ in their projections of the pattern and time-evolution of ENSO SST variability and El Niño and La Niña event evolution.

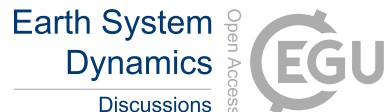

(c) The MEM projects an increase in ENSO SST variability, El Niño-like warming, and more variability in the central Pacific.

(d) The seasonality of ENSO SST variability increases. ENSO SST variability increases in boreal fall and winter with a decrease in spring and early summer. This is qualitatively consistent across all models.

2. Mean-state projections

(a) Most models project El Niño-like warming, although some models project the opposite.

(b) Individual models are different in their longitude and season of maximum warming in the tropical Pacific.

3. Relationship between ENSO SST variability and the tropical Pacific SST gradient

(a) When considering individual ensemble members, tropical Pacific SST gradient changes are linearly related to changes in ENSO SST variability in most models.

(b) This response is time-dependent. In many but not all models, ENSO variability first increases, then the tropical Pacific SST gradient weakens as ENSO variability plateaus or decreases.

4. A single forcing perspective

(a) Increases in ENSO SST variability in the two single forced SMILEs result from anthropogenic (aerosol and greenhouse gas) forcing.

(b) More single forced SMILEs are needed to understand the tropical Pacific SST gradient change.

These results present an extensive picture of future changes in ENSO in 14 SMILEs. The use of SMILEs means that the diverse responses across models can be truly attributed to model differences, rather than including contributions from internal variability. These diverse responses demonstrate a need for further investigation into the processes causing model differences in ENSO projections and provide a baseline for future research. Additionally, our results highlight time-dependent behaviour including non-linear changes in ENSO SST variability and changes in the tropical Pacific SST gradient that intensify near the end of the 21st century. This has important implications for ENSO's teleconnections and impacts, as a non-linear change in ENSO SST variability likely has non-linear time-dependent changes in its impacts as well. Our results may also help to reconcile previous work that suggests a transient increase in ENSO SST variability, but a long-term equilibrated decrease by isolating the time-dependent behaviour of ENSO SST variability projections. While many models agree on the trajectory of projected changes, not all models behave the same way. This highlights the need for further research on the mechanisms of inter-model differences in ENSO projections. There is a rich diversity of future ENSO changes projected by climate models and more work is needed to understand which aspects of these projections are robust

*Data availability.* Large ensemble data is available as follows:



– CanESM2, CESM1-LE, CSIRO-Mk36 & GFDL-CM3 are available from the multi-model large ensemble archive https://www.cesm.ucar.edu/projects/community-projects/MMLEA/

– ACCESS-ESM1-5, CanESM5, EC-EARTH3, IPSL-CM6-LR, MIROC6 & MIROC-ES2L are available from the CMIP6 archive at https://esgf-node.llnl.gov/projects/cmip6/

– CESM2-LE is available at https://www.cesm.ucar.edu/projects/community-projects/LENS2/

– GFDL-SPEAR-MED is available at https://www.gfdl.noaa.gov/spear_large_ensembles/

– GFDL-ESM2M data was provided by Prof. Thomas Frölicher at the University of Bern

– MPI-GE is available at https://mpimet.mpg.de/en/grand-ensemble/

*Author contributions.* This paper is a result of the ENSO in large ensembles workshop held at CU Boulder in August 2021. All authors
contributed to the analysis of data and conception of the paper at the workshop. NM wrote the manuscript with contributions from all authors. RCJW made Figures 7 & 9. JK made Figure 10. SM and MFS made Figures 1, 4 & 5. SCS made Figures 2 & S3. SS made Figure 8. XW made Figures 3, 6, S1, S2, S4, S5, S6 & S7.

*Competing interests.* The authors declare no competing interests

*Acknowledgements.* This paper is a result of the ENSO in large ensembles workshop held at CU Boulder in August 2021. NM was par-
tially funded by NSF AGS 1554659 in part by the CIRES Visiting Fellows Program and the NOAA Cooperative Agreement with CIRES, NA17OAR4320101. R.C.J.W. acknowledges support from the National Science Foundation (Grant AGS-1929775) and computing support from the Computational and Information Systems Laboratory at the National Center for Atmospheric Research (Casper Data Analysis Cluster; Project UWAS0094). M.F.S. was supported by NSF grant AGS-2141728, NOAA's Climate Program Office's Modeling, Analysis, Predictions, and Projections (MAPP) program grant NA20OAR4310445, and NOAA's Ocean Acidification Program (OAP) grant
NA21OAR0170191. SS was supported by the U.S. Department of Energy, DE-SC0019418 and by an NSF CAREER award, OCE-2142953. This is IPRC publication X and SOEST contribution Y (numbers updated on acceptance). X. Wu is supported by an Advanced Study Program postdoctoral fellowship from the National Center for Atmospheric Research (NCAR). S.C.S is supported by the University of Colorado Chancellor's Postdoctoral Fellowship program. CMIP6 data were obtained from the CMIP6 next-generation archive at ETH Zurich (Brunner et al. 2020). We acknowledge the World Climate Research Programme, which, through its Working Group on Coupled Modelling, coordi-
nated and promoted CMIP5 and CMIP6. We thank the climate modeling groups for producing and making available their model output, the Earth System Grid Federation (ESGF) for archiving the data and providing access, and the multiple funding agencies who support CMIP5, CMIP6, and ESGF.





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





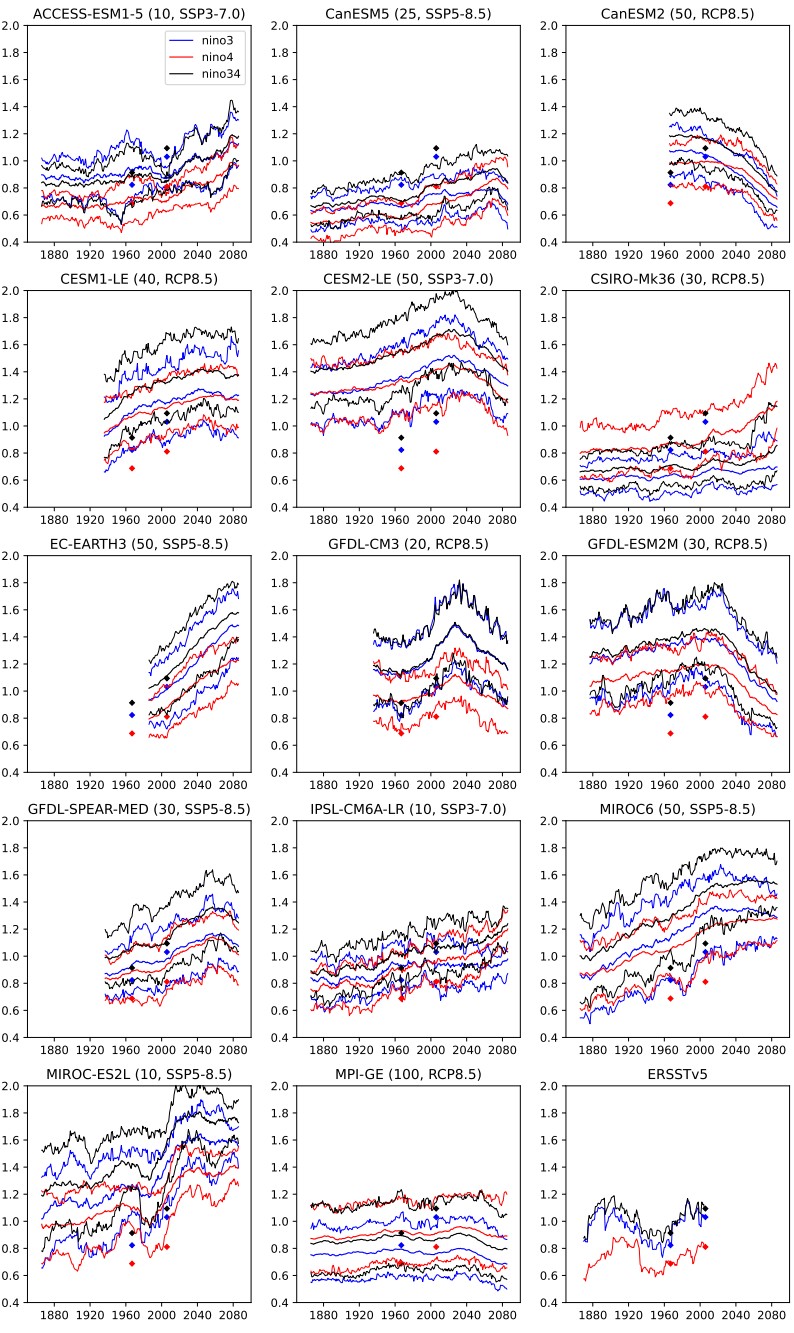

**Figure 1.** Time series of Niño3 (blue), Niño3.4 (green), and Niño4 (red) variability for the December, January, Feburary average (DJF) in each model. Variability is computed as the running 30-year standard deviation for each ensemble member after detrending each member by removing the ensemble mean. The solid line shows the ensemble mean of the running standard deviation, the thin dotted lines show the 5-95% range across the ensemble. The bottom right panel shows the running standard deviation for ERSSTv5 observations. The diamond symbols in each panel show the observed variability for the periods 1951–1980 and 1990–2019. Model name, ensemble size and scenario used are shown in the title of each panel for all Figures.



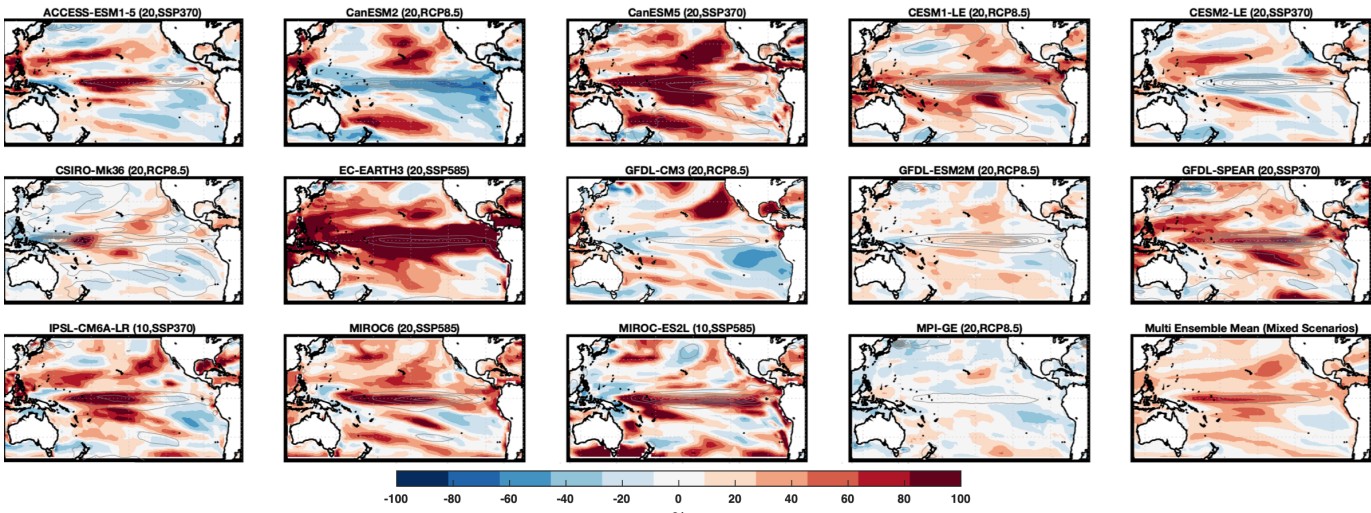

**Figure 2.** Spatial pattern of relative change in DJF SST variance in the (2-7 yr) ENSO band between the late 21st century(2070-2099) simulations and the historical (1951-1980) period. The difference in variances have been normalized by the variance of the historical (1951-1980) period (contours). The multi ensemble mean of the individual model ensemble means can be found in the last panel. Red colors indicate heightened variability, cool colors indicate dampened variability. For this and following Figures using DJF, 30 years are used in the calculations, e.g. for 1951-1980 we include 30 DJFs starting with December 1950 and DJF variance is computed as the variance of the DJF means. We note that a maximum ensemble size of the first 20 members is used in this analysis, with all members of models with 20 or less members used.



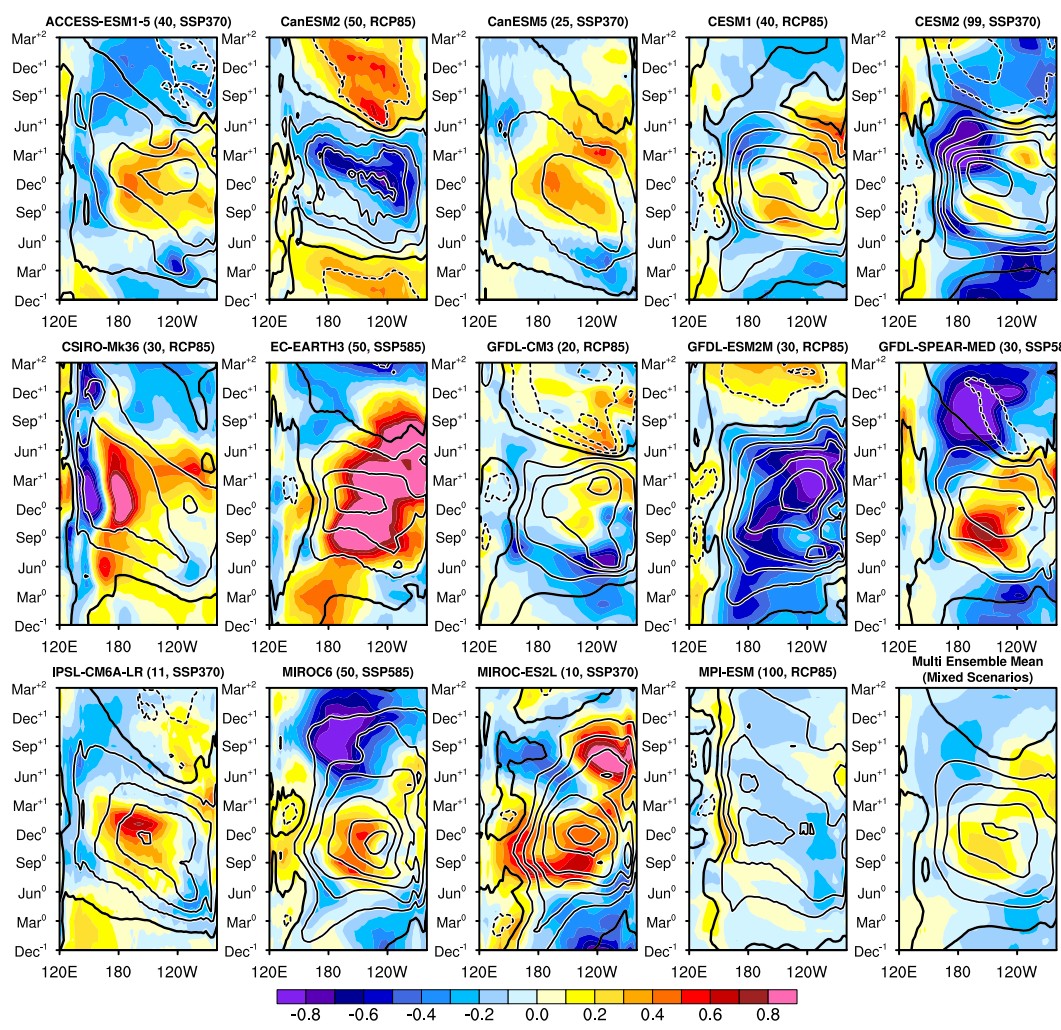

**Figure 3.** Longitude-time sections of the difference of equatorial Pacific (5°S-5°N) SST anomalies (°C; shading) composited for El Niño events during 2070-2099 compared to 1951-1980 in 14 SMILEs and the multi-ensemble mean. The SST anomalies for El Niño events during 1951-1980 are overlaid (°C; contours at intervals of 0.4°C; zero contours thickened and negative contours dashed). El Niño events are defined when the Niño3.4 index exceeds 0.75 standard deviations in Dec0 but not in Dec-1, where the standard deviation of Niño3.4 is calculated separately for each calendar month from October to February.



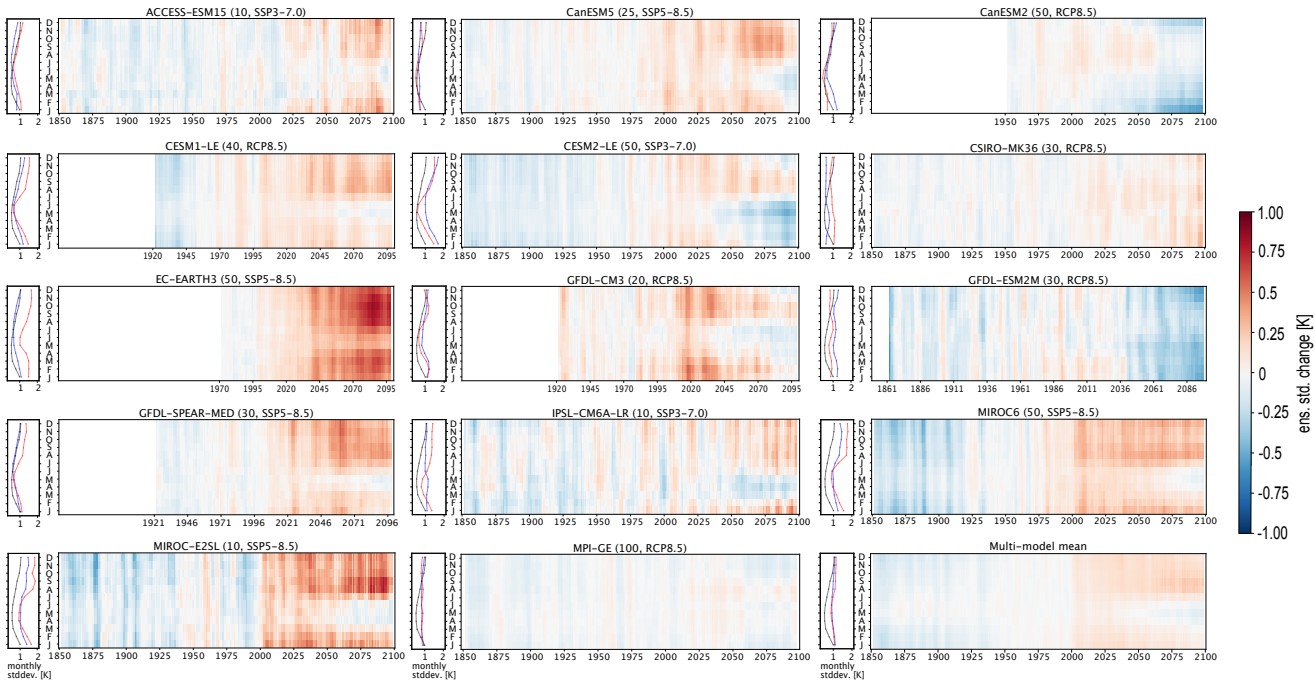

**Figure 4.** Change in ENSO SST variability, taken as the ensemble standard deviation in the nino3.4 region for each large ensemble and the multi ensemble mean. Blue lines show the climatology in 1951-1980, shading shows the 5-year-running-mean anomalies with respect to 1951-1980, and red lines show the climatology in 2070-2099. Black lines show the observational climatology based on ERSST5 in the period 1951-1980. We note that the climatology is computed as 1970-1999 for EC-Earth in blue and will be updated upon revision.



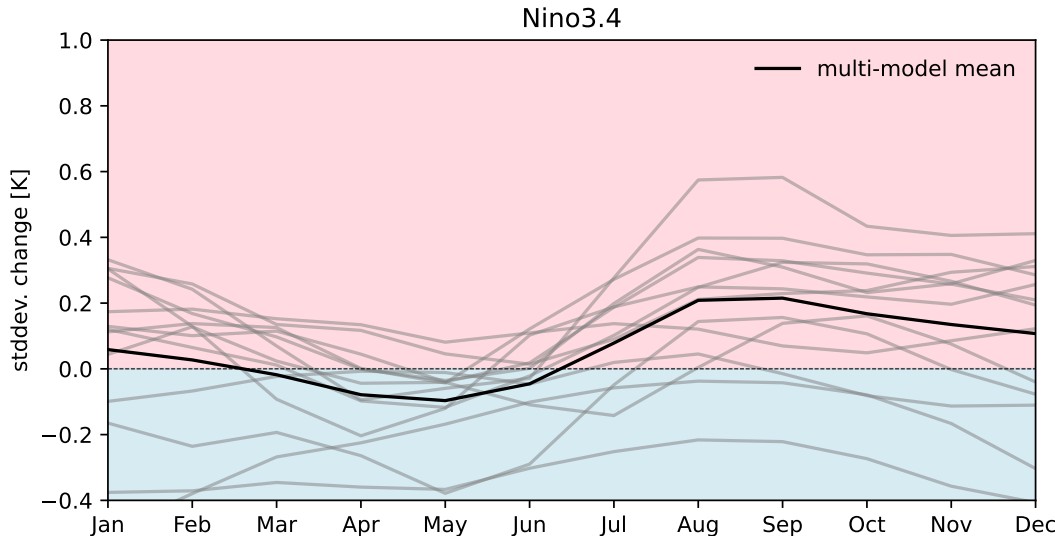

**Figure 5.** Change in ENSO SST variability taken as the ensemble standard deviation in the nino3.4 region for the period 2070-2099 as compared to 1951-1980. The grey lines are individual large ensembles, while the black line is the multi ensemble mean. Red shading indicates an increase in this metric, while blue indicates a decrease.

**Table 1.** SMILEs used in this study. CMIP6 models are highlighted in bold font.

| Model | Time period | Forcing (ensemble size) | Reference |
|---|---|---|---|
| **ACCESS-ESM1-5** | 1850-2100 | hist(40), ssp370(40) | Ziehn et al. (2020) |
| CanESM2 | 1950-2100 | Hist+rcp85(50) | Kirchmeier-Young et al. (2017) |
| **CanESM5** | 1850-2100 | hist(40), ssp370(25), | Swart et al. (2019) |
| | | hist-aer(15), hist-nat(8), hist-GHG(8) | |
| CESM1-LE | 1920-2100 | hist+rcp85(40) | Kay et al. (2015) |
| **CESM2-LE** | 1850-2100 | hist(100), ssp370(99) | Rodgers et al. (2021) |
| CSIRO-Mk36 | 1850-2100 | hist+rcp85(30) | Jeffrey et al. (2012) |
| **EC-EARTH3** | 1850(1970)-2100 | hist(1850; 23), hist(1970; 50), ssp585(58) | Döscher et al. (2022); Wyser et al. (2021) |
| GFDL-CM3 | 1920-2100 | hist+rcp85(20) | Sun et al. (2018) |
| GFDL-ESM2M | 1861-2100 | hist+rcp85(30) | Burger et al. (In review 2021) |
| **GFDL-SPEAR-MED** | 1921-2100 | hist+ssp585(30) | Delworth et al. (2020) |
| **IPSL-CM6A-LR** | 1850-2100 | hist(32), ssp370(10) | Boucher et al. (2020) |
| **MIROC6** | 1850-2100 | hist(50),ssp585(50), | Tatebe et al. (2019) |
| | | hist-nat(50), hist-aer(9), hist-GHG(3) | |
| **MIROC-ES2L** | 1850-2100 | hist(30), ssp585(10) | Hajima et al. (2020) |
| MPI-GE | 1850-2099 | hist+rcp85(100) | Maher et al. (2019) |



**Figure 6.** Longitude-time sections of the ensemble-mean linear trend of SST over the equatorial Pacific (5°S-5°N) during 2015-2099 normalized by global mean SST trend. The bottom right panel shows the multi-ensemble-mean climatological seasonal cycle of SST in the equatorial Pacific over 1950-1979. Vertical black lines delineate the averaging regions used in Figures 7-10 with the solid lines indicating the western Pacific region and the dashed lines indicating the eastern Pacific region.



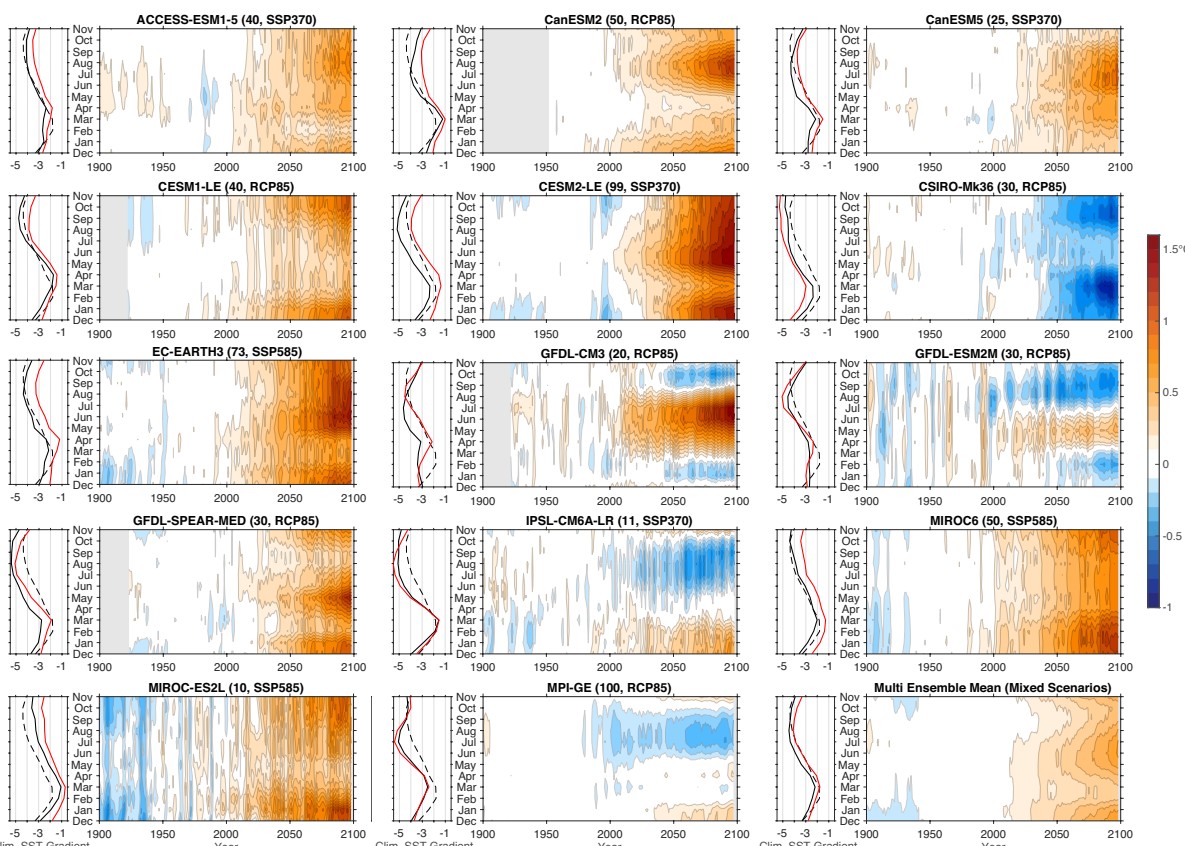

**Figure 7.** Tropical Pacific SST gradient climatology and mean-state changes, taken as the ensemble-mean difference between the eastern equatorial Pacific (90–150W, 5S–5N) and the western equatorial Pacific (120E-180, 5S–5N) for each large ensemble and the multi ensemble mean. Black lines show the climatology in 1951-1980, shading shows the 5-year-running-mean anomalies with respect to 1951-1980, and red lines show the climatology in 2070-2099. Black dashed lines show the observational climatology based on ERSST5 in the period 1951-1980. Red anomalies show El Niño-like changes and blue anomalies show La Niña-like changes.







**Figure 8.** Time differences of the Niño3.4 standard deviation (y-axis) and the difference in SST between the east and west equatorial Pacific (defined as in Figure 7; x-axis) for each ensemble member of the 14 ensembles, indicated by colored symbols. a) Difference between 2021-2050 and 1950-1980; b) Difference between 2071-2099 and 1950-1980. Niño3.4 time series are detrended by subtraction of the time-varying ensemble mean prior to computation of standard deviation.



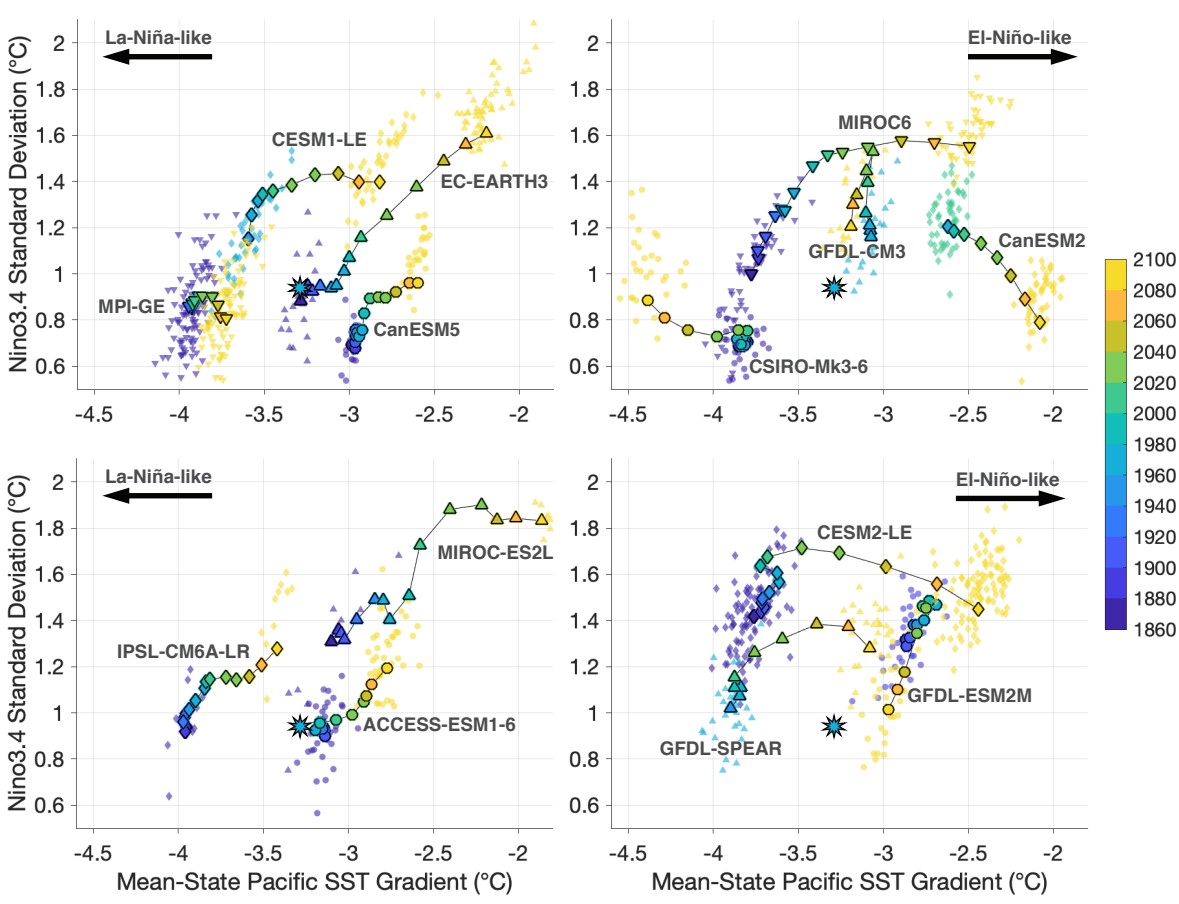

**Figure 9.** Time evolution of the 30-year-running-mean DJF tropical Pacific SST gradient (defined as in Figure 7; x-axis) and 30-year running standard deviation of DJF Niño3.4 variance (y-axis) in each SMILE. Symbols with black edges show the ensemble mean for overlapping 30 year periods every 15 years between 1850 and 2100, where the center year of the averaging period is computed is indicated by the fill color. The observational values for ERSST5 during the period 1951-1980 are shown with a light blue star. The 30-year -running-mean DJF tropical Pacific SST gradient and 30-year running standard deviation of DJF Niño3.4 in each ensemble member are shown for the first and last averaging periods (symbols without black edges).



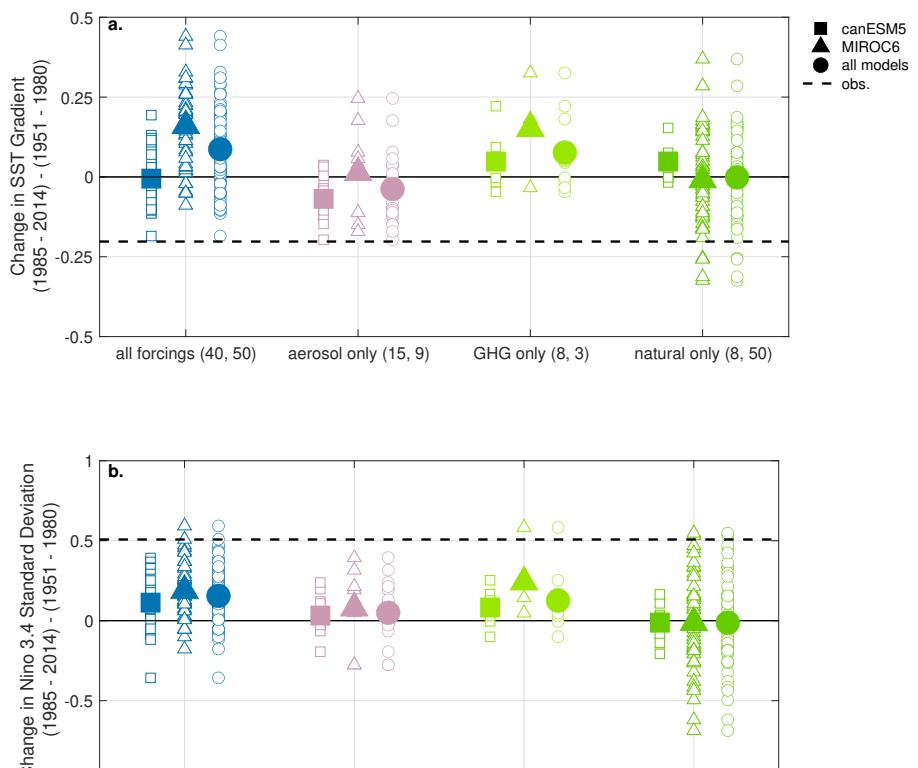

**Figure 10.** Change in the tropical Pacific SST gradient and Niño 3.4 index standard deviation in DJF in the two available single-forcing ensembles. (a) Change in the tropical Pacific SST gradient (west - east; defined as in Figure 7) over the second half of the 20th century (1985 - 2014 minus 1951 - 1980). Individual ensemble members from CanESM5 (squares), MIROC6 (triangles), and the combination of the two ensembles (circles) are shown with open symbols. The average change in gradient is shown with a filled marker. Observed changes (ERSSTv5) are shown in the dashed black line. (b) Change in the Niño3.4 index standard deviation over the same time period. We use the same visual conventions. Note that the sum of individual forcings may not add to the all-forcings, particularly in relatively small ensembles.