# Peer review of "The future of the El Niño-Southern Oscillation: Using large ensembles to illuminate time-varying responses and inter-model differences"

_Earth System Dynamics, 2022_

## Author Response (AR1)

**Review 1**

Review of "The future of the El Nino-Southern Oscillation: Using large ensembles to illuminate time-varying responses and inter-model differences"

I find the paper to be informative and I believe that the community would be interested in this work. The paper is generally well written. My detailed comments are listed below.

*We thank the reviewer for their time, positive review and helpful comments.*

   • Line 19 P2, add Cai et 2012 Nature https://www.nature.com/articles/nature11358; 2014 NCC https://www.nature.com/articles/nclimate2100#citeas, as these are among the earliest papers on the topics?

*Citations have been added on line 19*

   • Line 23, the difference between Cai et al., 2022 and [Wengel et al., 202, Callahan et al., 2021] lies in that one is transient and the others are stabilised CO2. This should be clarified so as not to create further confusion. Line 45 seems to reinforce the confusion.

*We have removed the citation to Wengel at al on line 23. Line 45 (now 42) has been modified to read:*

 *"While consensus between multi-model ensembles and paleoclimate records suggests that ENSO variability may increase with increasing greenhouse gas emissions, other studies that look at the equilibration to strong greenhouse gas forcing complicate this result."*

   • Lines 55-60, paleoclimatic proxy suggests that there is no relationship between mean zonal SST gradient and ENSO variability (Cai et al 2021).

*The following is added on line 64:*

 *"However, paleoclimatic proxies suggest that there is no relationship between mean zonal SST gradient and ENSO variability (Cai et al, 2021), demonstrating that this link is either not found in our observations or, as in the PMIP models, is not constant in time. "*

   • Line 93, please cite a butterfly effect paper
https://www.nature.com/articles/s41586-020-2641-x as it is easy to understand. I think the paper also suggests that there is an effect on future ENSO evolution from the initial period.

*While this is an interesting paper line 93 references the comparison of CMIP spread to the spread of a single model for which the references Maher et al, 2018 and Ng et al 2021 are more relevant. As such we choose to leave the citation as is.*

• Line 155 onward, it is not clear if anomalies are constructed referenced to climatology of individual experiment or the ensemble mean. It should be the former. By definition, a climatology is the average of all years that contribute, such that the anomalies sum to zero. If it is the latter, then the inter-experiment difference in climatology needs to be assessed, and the anomalies might not sum to zero.

*We remove the ensemble mean. This is because we aim to construct anomalies where the forced signal is removed. The forced signal is well estimated by the ensemble mean (see https://agupubs.onlinelibrary.wiley.com/doi/full/10.1029/2019MS001639 and https://www.nature.com/articles/s41558-020-0731-2 for details on use of ensemble mean to estimate the forced response).*

*We have made this methodology clear in the Figure caption by adding:*

*"Monthly SST anomalies are calculated by removing the ensemble mean from individual ensemble members."*

• Lines 166-167, "CESM2 is an exception that has opposite changes in El Nino SST amplitude and La Nina duration between the two periods." The Cai et al. 2020 seems to provide a mechanism for this?

*This result is not directly comparable to Cai et al, 2020 (assuming this refers to the butterfly effect paper cited above) which looks at the evolution of individual ensemble members, while we consider the ensemble mean as an estimate of the forced signal. If we were considering how individual ensemble members evolved this paper by Cai et al, would be extremely relevant. However, as we consider the forced response on the ensemble as a whole rather than individual model trajectories this work does not explain the mechanism for CESM2 having opposite changes between 2 periods.*

• Figure 1, it is interesting that for a SMILE, most experiments behave in a similar way, either unidirectional or reversing, suggesting that it is strongly model dependent. What causes the dependence?

*Given a SMILE consists of experiments that are all run with the same model, it makes sense that the individual simulations will have a similar overall trajectory. However, we note that individual experiments are not shown in Figure 1 – what we show is the ensemble mean and the 5-95% range across the ensemble. This means that the internal variability or trajectory of the individual members within the ensemble spread is not illustrated. The individual members while following the same overall trajectory will show much more noise than the ensemble mean and spread.*

*Model dependence could be related to the following:*
      *-Different climatological biases*

*- Different patterns of transient mean-state warming*
*-Different ENSO feedbacks and dynamics*

*See following references:*
*Planton et al, 2021:*
*https://journals.ametsoc.org/view/journals/bams/102/2/BAMS-D-19-0337.1.xml*
*Bellenger et al 2014: https://link.springer.com/article/10.1007/s00382-013-1783-z*
*Wills et al, 2022: https://agupubs.onlinelibrary.wiley.com/doi/pdf/10.1029/2022GL100011*
*Capotondi et al, 2015: https://repository.library.noaa.gov/view/noaa/31041*

*We have added the following on line 284*

*" Model differences could be related to different climatological biases, different patterns of transient mean-state warming and different ENSO feedbacks and dynamics (e.g. Bellenger et al., 2014; Capotondi et al.,2015; Planton et al., 2021; Wills et al., submitted 2022) with additional work needed to diagnose why individual models behave the way they do."*

• Line 175, what is the dynamics for increased ENSO seasonality?

*We cannot answer this without a full feedback analysis, which is out of the scope of this study. We have added the following text on line 293*

*"Determining the dynamical cause for the increased ENSO seasonal synchronization in most of the models will require a detailed ENSO feedback analysis (e.g., Chen & Jin 2022), including assessing potential future changes in the "southward wind shift" mechanism (e.g., McGregor et al. 2012, Stuecker et al. 2013)."*

• Line 220, are you able to further test the idea of nonlinearity controlling mean state change by relating them in an inter-model/experiment relationship?

*We have updated the text on line 220 (now 239) to read:*

*"The internal variability relationship that is the primary contribution to Figure 8a clearly shows the role of rectification into the mean state (e.g. Hayashi et al., 2020), but for the forced changes this is only one of several mechanisms going on, so the forced changes can depart from this linear relationship (Figures 8b and 9)."*

• Lines 229 and 289, what is the dynamics for increasing aerosols to drive an increase in ENSO variability? One would expect increasing aerosols to have an opposite impact to that of increasing CO2. Is it possible that internal variability plays a role in the result?

*This is currently unresolved. Aerosol forcing, however, is not the simple inverse of CO2 forcing as it is hemispherically asymmetric unlike greenhouse gas forcing. This leads to ITCZ shifts that can influence ENSO.*

*See following references:*
*Luongo et al, submitted: https://www.essoar.org/pdfjs/10.1002/essoar.10512160.1*
*Kang et al 2020: https://www.science.org/doi/10.1126/sciadv.abd3021*
*For stratospheric volcanic aerosol:*
*Pausata et al , 2020: https://www.science.org/doi/10.1126/sciadv.aaz5006*

*We have added the following on line 320*
*"While one might expect aerosol forcing to have the opposite effect of greenhouse gas increases, aerosols are hemispherically asymmetric unlike greenhouse gas forcing leading to shifts in the intertropical convergence zone that can also influence ENSO (Kang et al., 2020; Pausata et al., 2020)."*

**Review 2**

This paper examines changes in ENSO SST anomalies in a number of large ensembles. It is really a 'show and tell', looking at changing SST variability using a number of different measures. There is a significant amount of data wrangling involved in this type of work and the authors are world leading in this regard. The analysis is approached in a careful way and it supports the conclusions of the paper. Figures and text are of high quality.

*We thank the reviewer for their time taken to review the paper and positive review.*

Perhaps the most dissapointing thing, however, is that there is little insight provided as to why the large ensembles behave in such diverse ways. Some show increases, some decreases and some show non-linear responses in variability. Understanding this latter behaviour would be of significance scientific interest to the ENSO/climate change community. There are simple metrics available to look at mechanistics aspects of ENSO changes in models and it is a bit of a shame that the authors do not try some of these e.g. assessing the atmos-ocean coupling strength and its components. Such an analysis would significantly enhance the work.

*This paper was a community effort that came out of a small workshop, and many subsets of the authors are now pursuing follow-up work to better understand the mechanisms of the responses identified in this work, which necessarily makes up multiple additional projects/papers due to the wide range of topics involved (e.g., non-linear ENSO amplitude responses, ENSO seasonality changes, ENSO response to aerosol forcing, inter-model diversity in mean-state gradient changes, etc.). The aim of our current study is to illuminate how ENSO behaves in all available large ensembles. Here, we can look at ENSO evolution over time due to the use of large ensembles and truly identify how each model behaves under a strong warming scenario. We agree that understanding why the models behave differently is an important question. However, this is out of the scope of our study, which already includes 10 Figures in the main text and 7 in*

*the Supplementary. We hope this work will inspire others (as well as subsets of authors already working on this topic) to look further into these new datasets.*

*We have additionally already added references with more hypotheses of mechanisms in response to Reviewer 1. See the following examples:*

*Line 284 -*
*" Model differences could be related to different climatological biases, different patterns of transient mean-state warming and different ENSO feedbacks and dynamics (e.g. Bellenger et al., 2014; Capotondi et al.,2015; Planton et al., 2021; Wills et al., submitted 2022) with additional work needed to diagnose why individual models behave the way they do."*

*Line 294 -*
*"Determining the dynamical cause for the increased ENSO seasonal synchronization in most of the models will require a detailed ENSO feedback analysis (e.g., Chen & Jin 2022), including assessing potential future changes in the "southward wind shift" mechanism (e.g., McGregor et al. 2012, Stuecker et al. 2013)."*

*Line 239 -*
*"The internal variability relationship that is the primary contribution to Figure 8a clearly shows the role of rectification into the mean state (e.g. Hayashi et al., 2020), but for the forced changes this is only one of several mechanisms going on, so the forced changes can depart from this linear relationship (Figures 8b and 9)."*

It also seems a little odd that the authors do not make some comments on minimum ensemble size for looking at changes in ENSO.

*This is a good point.*

*The following discussion is added on line 111*

*"The 14 SMILEs used in this study include both CMIP5 and CMIP6 class models and use one of three external forcing scenarios (Table 1). Models used in this study have between 10 and 100 ensemble members for the historical and future scenarios. For those with single forcing, we allow as few as 3 members due to the limited availability of models with any more than a single ensemble member. Previous work has investigated how many ensemble members are needed to investigate ENSO characteristics and find that this depends on the metric used, the length of time averaging, the level of acceptable error and the model itself (Maher et al., 2018; Milinski et al., 2020; Lee et al., 2021). Lee et al. (2021) find, for example, that for ENSO amplitude 10-47 members are sufficient depending on which model is used to make the estimate, while this range widens to 1-45 members for ENSO seasonality. Maher et al. (2018) determine that to look at ENSO variance over 30-year periods 10 members are sufficient for a 20% level of acceptable error, while 30-40 are needed to reduce this to 10%, with 12-20 sufficient for a 15% error*

*(Milinski et al., 2020). Less members are needed to estimate mean state ENSO metrics, than its variability (Milinski et al., 2020; Lee et al., 2021). In this study we accept models with a minimum of 10 ensemble members as to include as many models as possible, however, we note that there are larger errors associated with estimates of ENSO variability for the smaller ensemble sizes*

**Review – John Fasullo**

Review of The future of the El Niño-Southern Oscillation: Using large ensembles to illuminate time-varying responses and inter-model differences
by Maher et al.

The manuscript by Maher et al. seeks to diagnose changes in ENSO in 14 single model large ensembles (so-called SMILEs). The manuscript builds upon a body of work that is often based on single members, individual models, or idealized models and so it represents an advance, particularly at resolving the decadally varying aspects of forced changes in variance - which the work shows can be important. The manuscript is clearly written, is explicit about its objectives, findings, and reasoning, and includes figures that are well-designed and clear. There is sufficient new material here to justify publication. Some aspects are frustrating - such as in cases where the robust take-home message seems to be that there is no robust-take home message. Though basic questions go unanswered concerning the origins of inter-model contrast and mechanisms of change, the broader community has also struggled to answer these questions and so this work is not unique in this regard. That said I do have some minor suggestions for improvement. This includes the general suggestion that multi-model means not be used for many of the metrics because of the disproportionate variance in some models that is swamping out the means. Rather I think medians make more sense since the broader goal is to make generalizations about model behavior, which implicitly seeks to screen out outliers. I have various other relatively minor suggestions listed below but otherwise view the manuscript as suitable for publication.

*Thank you for taking the time to review this paper and for your positive comments and helpful suggestions.*

*While we agree that the median would make more sense for broader generalizations, however, we choose to use the multi ensemble mean (MEM) for easy comparison with previous work that uses multi-model means (MMM) from CMIP. Here, the main aim is to show the diversity of forced responses that go into such a MMM and demonstrate that these diverse individual model responses average out in the MEM to show something that looks very similar to the MMM from previous studies. As such we choose to use the MEM rather than medians in this study. We have addded a sentence on this point in the revised manuscript to make the use of MEM clear to the reader.*

*Line 159-*
*"multi-ensemble mean (MEM; used for ease of comparison to CMIP multi-model mean results) "*

\*\*\*

55: There have been various studies that show the improvement of ENSO simulation across CMIP generations. These seem useful to reference in this paragraph to provide context on the numerous SMILEs.

*We have revised line 125 to read "A detailed comparison of ENSO characteristics in both CMIP models and SMILEs can be found in the following studies for a large range of ENSO metrics, and we do not repeat this here, although we note that in general CMIP6 models outperform CMIP5 in 8 out of 24 metrics and are only degraded in one (Planton et al., 2021; Lee et al., 2021) and that ENSO related SST biases and its teleconnections are also improved (Fasullo et al., 2020)."*

119: I appreciate that there can be useful examples of models that suggest that no simple relationship between the present day and future exists but perhaps a more thorough exploration across all models and metrics considered here would be move convincing?

*We have added a scatter plot considering this question as Figure S1. We have also added the following text:*

*Line 134 - "When considering ENSO SST variability we find that only some models capture the observed variability within the ensemble spread (Figure 1 S1), but note that this is not a reliable predictor for future change."*

*Line 211 - "With the exception of CSIRO-Mk36, models with a stronger climatological gradient tend to have stronger El-Niño-like warming (Figure S1b), though the correlation between the gradient climatology and the gradient change is not significant."*

126: It is worth calling out the very large differences between SSP370 and RCP85/SSP585 in regarding to early 21st sulfate aerosols and potential consequences for the evolution of ENSO.

*Thanks for the comment –we have added the following text on line 143:*

*"We note that there may be differences between SSP370 and RCP85/SSP585 particularly regarding to early 21st sulfate aerosols which has potential consequences for the evolution of ENSO (O'Neill et al.,2014; Fasullo and Richter, 2022)."*

Figure 1: There is substantial white space in the 3x5 layout of the figure. I recommend changing to 4x4 and ensuring there is little white space as the figure is somewhat inefficient and difficult to read as is. Also it looks as thought the vertical extent needs to be expanded as some lines go out of range. Note also that there is substantial noise in many of the time series. I suggest applying a smoother except where the variability is not irrelevant noise. That said, various models seem to have abrupt responses to forcing, such as volcanic eruptions and perhaps even

biomass effects (in MIROC-ES2L), that is too abrupt to be explained by warming alone. The authors seem not to find this worthy of discussion? I recommend addressing it.

*We keep the time-series as is acts to demonstrate the inherent variability due to different ensemble sizes. Given the ensemble sizes are different this is helpful to give an idea as to how noisy the results we have are. We have revised the vertical extent in the revision and optimised the white space.*

*We agree the abrupt changes are interesting, however they are difficult to diagnose without a thorough analysis of each models individual forcing differences. They may also be artifacts of the smaller ensemble size of models such as MIROC-ES2L.*

*On line 123 we add " We additionally show the timeseries of ENSO indicies in Figure1 as is acts to demonstrate the inherent variability due to different ensemble sizes. "*

Figure 3: The multi-ensemble mean seems to be dominated by EC-Earth3. Would a multi-ensemble median perhaps be more appropriate?

*See comment at the beginning of the response.*

Figure 4: Fonts are too small - minimum font including axes should be on par with main text. Monthly stddev lines should be make thicker.

*This has been revised as suggested.*

The authors don't provide any hypotheses for the seasonality of the change in variance? Do none exist? Again the MIROC-ES2L increase at 2000 is quite notable. Is there no explanation for why this may occur? Other models show periodic changes in variability in the future. Is this just noise? Does it suggest that again some additional smoothing is needed to deal with some small ensemble sizes using monthly data? If one looks at CESM2 there are again suggestions of periodicity? What might drive this? I suggest reducing the range of the color bar to make colors in the figure more visible.

*We have reduced the colorbar range. In terms of the seasonality we cannot answer this without a full feedback analysis, which is out of the scope of this study. We have added the following text on line 293*

*"We note that determining the dynamical cause for the increased ENSO seasonal synchronization in most of the models will require a detailed ENSO feedback analysis (e.g., Chen & Jin 2022), including assessing potential future changes in the "southward wind shift" mechanism (e.g., McGregor et al. 2012, Stuecker et al. 2013). "*

Figure 5: Perhaps put whiskers on Fig 5 corresponding to the 2 standard error range, which seems to increases in the latter half of the year? Might multi-model median be more appropriate due to sensitivity of means to a single model?

*We have added another line for the median in this plot. Rather than add a error range we choose to show each individual model in light grey to illustrate what goes into the multi-model mean.*

Why are 99 members of CESM2 used for some figures and only 50 used for others? This is not discussed at all and contradicts with Table 1. I'd review all plots and ensure that the # of members used is consistent with Table 1.

*Thanks for pointing this out – we have made sure all 99 are used where there we only 50 in the revised version.*

Figure 6: I again question the use of multi model mean rather than median given the outsized influence of some models (CSIRO).

*See comment at the beginning of the response.*

Figure 8: Since the abscissa is not symmetric I recommend that a vertical line at 0 be shown to avoid confusion. As is, the panels and particularly the top, are misleading.

*This has been added into the revised version.*

Figure 8: Doesn't the fact that obs have become more La Niña like suggest that we should have seen a reduction in variance?

*If you look at individual members you can see that while it is more likely with a La-Nina like change that there is a reduction in variance, not all members do this – showing the inherent noise in the system. The addition of 0 lines as suggested above have made this clearer.*

Figure 9: There should probably two more sets of arrows at the top of the plot saying La Niña-like and El Niño-like such that they are on each plot.

*We have added these.*

226: Why not also infer changes from the CESM1-SF (and now CESM2-SF if you do add CESM1 to the figure)?

*Using the difference between the all and the all-but forcing used in CESM1-SF does not necessarily give the same result as a single forcing experiment. For consistency, we only use single-forcing not all-but forcing experiments. CESM2-SF was not available when the workshop where the data analysis was done for this publication was held.*

Figure 10: reference to 'all models' is a bit confusing given there are only 2 models. perhaps state "both models" or better yet "both ensembles"? or one could include ALL CMIP6 DAMIP simulations to provide context for canesm5 and miroct6. Also I wonder why not show these in a similar fashion to Fig 9?

*We cannot find the reference to all models in this section that is referenced by the reviewer.*

*We do not show these the same way as in Figure 9 as the periods we are looking at are much shorter and the single forcing ensemble sizes are inconsistent within the same model. Some of the small ensemble sizes made it difficult to understand the ensemble mean – this is why we show each member as well as the mean in Figure 10.*

255: The discussions based on MEM should be reconsidered in the context of the median I think, particularly when making categorical statements of models overall (since the mean is strongly weighted by only a few models with large variance and is therefore not representative).

*See response at the beginning of this document.*

I recommend that the paper include a discussion of possible mechanisms and paths forward for exploring.

It is unfortunate that there is so little consistency across models on many aspects. The reader is left to wonder a bit on what is the robust finding here relevant to nature, aside from little being consistent across models? Have the authors looked for connections between some of the metrics being shown across models (e.g. seasonality of mean state changes and ENSO variance)? Have the authors examined what systematic differences exist for models that do best in some metrics packages (e.g. CVDP) in the present day (if so it would be go to mention, if not it would be good to do)? or for which changes in variance in observations fall within the ensemble spread, though this may be a weak constraint but would still be good to mention.

*The aim of our paper is to illuminate how ENSO behaves in all available large ensembles. Here, we can look at ENSO evolution over time due to the use of large ensembles and truly identify how each model behaves under a strong warming scenario. We agree that understanding why the models behave differently and trying to better understand the consistency between models is an important question. However, this is out of the scope of our study, which already includes 10 Figures in the main text and 7 in the Supplementary. We hope this work will inspire others to look further into these new datasets. Our final point in the manuscript aims to highlight this point and hopes to inspire further studies that look into this question in more detail (see line 354).*

*"This highlights the need for further research on the mechanisms of inter-model differences in ENSO projections. There is a rich diversity of future ENSO changes projected by climate models and more work is needed to understand which aspects of these projections are robust."*

*We have additionally already added references with more hypotheses of mechanisms in response to Reviewer 1. See the following examples:*

*Line 365 -*
*" Model differences could be related to different climatological biases, different patterns of transient mean-state warming and different ENSO feedbacks and dynamics (e.g. Bellenger et al., 2014; Capotondi et al.,2015; Planton et al., 2021; Wills et al., submitted 2022) with additional work needed to diagnose why individual models behave the way they do."*

*Line 294 -*
*"Determining the dynamical cause for the increased ENSO seasonal synchronization in most of the models will require a detailed ENSO feedback analysis (e.g., Chen & Jin 2022), including assessing potential future changes in the "southward wind shift" mechanism (e.g., McGregor et al. 2012, Stuecker et al. 2013)."*

*Line 239 -*
*"The internal variability relationship that is the primary contribution to Figure 8a clearly shows the role of rectification into the mean state (e.g. Hayashi et al., 2020), but for the forced changes this is only one of several mechanisms going on, so the forced changes can depart from this linear relationship (Figures 8b and 9)."*

304: Projections of nearly every climate quantity are nonlinear in time (since radiative forcing is nonlinear). Is there really an expectation of linearity? and if not is this really a significant result?

*While there is not an expectation of linearity, most previous studies have not been able to look at the time-dependent response to assess whether there is linearity. For example most previous studies have considered a future time period (e.g. 2000-2099) compared to a historical period (e.g. 1850-1950). Furthermore, the nonlinear responses go beyond the nonlinearity of the forcing to show non monotonicity in some models.*

308: As stated earlier I'd avoid the MEM(ean) and use the median.

*See comment at beginning of document.*

334: last sentence needs a period.

*Thanks, this is done.*

None of the panels in the paper are labeled (e.g. A, B, C, …). I think they should be for easier reference in this and other manuscripts that may cite specific figure panels from this work.

*We choose not to add these as they will make the Figures overly busy. Others can still reference the individual Figures and it should be easy to identify individual models as they are named in the titles.*

---

## Author Response (AR2)

**Response to reviewer 3**

Review of Maher et al., "The future of the El Niño-Southern Oscillation: Using large ensembles to illuminate time-varying responses and inter-model differences."

This manuscript clearly and effectively shows the effect of global warming on ENSO in an unprecedented set of model simulations, using well-established methods from large ensembles. The findings are well-explained, the figures are clear, and the analysis appears technically sound.

My basic impression from the first round of review is that I share reaction of other reviewers that the manuscript could go further in terms of physical insight and takeaways. The fact that the models differ but we don't really know why is ultimately somewhat unsatisfying. That being said, that is ultimately the state of the science on ENSO and not the authors' fault. Additionally, the forced responses shown in this paper could provide a basis for further investigation, making it a useful and important advance. For example, the non-monotonicity shown in Figure 1 is fascinating and, on its own, a worthwhile subject of research that I hope the authors (or others) are pursuing.

Therefore, I recommend this manuscript be published. I do have some very minor comments, but nothing that should strongly impede publication.

*We thank the reviewer for their positive review and helpful comments.*

Line numbers refer to the revised, non-tracked-changes manuscript.

The definitions of the Pacific SST gradient are inconsistent in places. In Figures 7 and 8, it's defined as "the difference between the eastern equatorial Pacific and the western equatorial Pacific" (Fig. 7) and "E-W Pacific SST difference" (Fig. 8), but in the Fig. 10 caption it says "west – east." This may be just be a typo, but I recommend making sure these definitions are consistent since it can get confusing.

*We apologize the Figure 10 caption was incorrect. It has now been updated.*

Personally, I'd prefer everything to be defined as west minus east rather than east minus west (for example, like this study defined the Walker circulation: doi:10.1038/nature04744). Positive values on the x-axis in Figure 8 refer to "El Niño-like warming," but that's really referring to decreases in the SST gradient, so it's counterintuitive for positive values to refer to a decrease in something. But either way, just make sure it's consistent and clearly defined.

*We choose to keep east minus west as noted by the reviwer this keeps the weakening as a less negative value rather than the counter-intuitive positive values. We have checked the manuscript to make sure everything is well and consistently defined.*

Line 41: A recent paper (https://doi.org/10.1038/s41467-023-36053-7) showed the contributions of both anthropogenic warming and multidecadal variability in shaping 20th-century ENSO changes and could be cited/discussed here. The strong contributions of natural variability apparent in their results further emphasize the need for large ensembles.

*We have added the citation on line 41 with the following text:*

*"A recent study also noted the strong contribution of internal variability in the observed record (Gan et al., 2023). Gan et al. (2023) conclude that 65% of the observed increase in extreme El Niño events is attributable to the internal variability (largely multi-decadal variability from the Atlantic Multidecadal Oscillation) with the rest of the increase attributed to a changing climate."*

Figure 10: I recommend switching the panels (make the change in ENSO variability the top panel), since you discuss the ENSO variability portion first (lines 252-259) and then move to the SST gradient later.

*This has been updated.*